# BioPlanner: Automatic Evaluation of LLMs on Protocol Planning in Biology

**Odhran O'Donoghue**[1,2,4]    **Aleksandar Shtedritski**[1,2,4]    **John Ginger**[1,2]
**Ralph Abboud**[1,2]    **Ali Essa Ghareeb**[2]    **Samuel G Rodriques**[2,3]

[1] Align to Innovate    [2]Francis Crick Institute    [3]Future House    [4]University of Oxford

## Abstract

The ability to automatically generate accurate protocols for scientific experiments would represent a major step towards the automation of science. Large Language Models (LLMs) have impressive capabilities on a wide range of tasks, such as question answering and the generation of coherent text and code. However, LLMs can struggle with multi-step problems and long-term planning, which are crucial for designing scientific experiments. Moreover, evaluation of the accuracy of scientific protocols is challenging, because experiments can be described correctly in many different ways, require expert knowledge to evaluate, and cannot usually be executed automatically. Here we present an automatic evaluation framework for the task of planning experimental protocols, and we introduce BIOPROT[1]: a dataset of biology protocols with corresponding pseudocode representations. To measure performance on generating scientific protocols, we use an LLM to convert a natural language protocol into pseudocode, and then evaluate an LLM's ability to reconstruct the pseudocode from a high-level description and a list of admissible pseudocode functions. We evaluate GPT-3 and GPT-4 on this task and explore their robustness. We externally validate the utility of pseudocode representations of text by generating accurate novel protocols using retrieved pseudocode, and we run a generated protocol successfully in our biological laboratory. Our framework is extensible to the evaluation and improvement of language model planning abilities in other areas of science or other areas that lack automatic evaluation.

## 1 Introduction

Traditional manual methods for research in biology are time-consuming, labour-intensive, and highly prone to human error. Robotic laboratory automation has the potential to increase accuracy, repro-

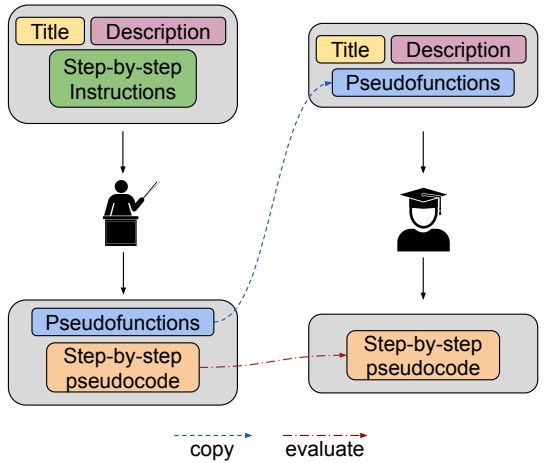

Figure 1: **Automatic evaluation of protocol generation.** The teacher model is given full information about a scientific experiment protocol – title, description, and step-by-step instructions. It is prompted to generate pseudo functions that allow the execution of the protocol. The student model is given the admissible pseudofunctions and is evaluated on its ability to generate the step-by-step pseudocode.

ducibility, and scalability, contributing to more scientific breakthroughs and a faster transition from research to real-world applications.

One important step towards automation of biology research is the automated generation of a laboratory protocol (Accurate step-by-step instructions on how to complete an experiment to accomplish a specific goal) which can subsequently be converted into robot code. LLMs have significant latent scientific knowledge and thus may be able to formulate accurate scientific protocols, which has been demonstrated for the field of chemistry (Bran et al., 2023; Boiko et al., 2023). However, to-date there has not been any clear way to evaluate the accuracy of a generated scientific protocol, except by manual evaluation. Without established evaluation metrics, progress in the field of automating science remains challenging.

Evaluating laboratory protocols is difficult for

---

[1]The dataset and code for evaluation are available at https://github.com/bioplanner/bioplanner

two reasons. Firstly, protocols are very sensitive to tiny details, and slight variations in instructions can lead to significantly different outcomes. When comparing generated protocols against ground truths, metrics that rely on n-gram overlaps such as BLEU (Papineni et al., 2002) or contextual embeddings such as BERTScore (Zhang et al., 2019) might not capture small differences, such as the order of actions, or relation between substances (Bhandari et al., 2020). Secondly, the same protocol can be described correctly at various levels of granularity. The same technique (e.g. sequencing library preparation) can be described by a single line or multiple paragraphs. This variability in granularity makes it difficult to evaluate the accuracy of LLM-generated protocols.

We here present an automated approach to evaluating the ability of a language model to write biological protocols. Our approach is inspired by robotic planning, in which a closed set of admissible actions is provided to a controller agent (Jiménez et al., 2019; Ahn et al., 2022; Huang et al., 2022). We use GPT-4 to automatically convert a written protocol into pseudocode using a protocol-specific set of pseudofunctions that is generated by the model (see Figure 1). Here, a "teacher" model generates the admissible action set and correct answer in terms of step-by-step pseudocode. Having access to this privileged information, we can then evaluate the performance of a "student", that has to solve the task from scratch. In this way, we then evaluate the ability of language models to generate a protocol when presented only with the appropriate pseudocode functions and a short description of the protocol. In effect, our approach allows us to automatically convert the process of writing a scientific protocol into a series of multiple-choice questions (i.e., pick a pseudofunction from a provided set), which can be evaluated much more robustly than natural language generation. This paradigm allows us to rapidly measure the protocol knowledge of GPT-3.5 and GPT-4 with minimal human intervention, and can serve as a general approach for evaluating and improving long-horizon planning in open-ended tasks in the future.

To this end, we also introduce a novel dataset, BIOPROT, of publicly available biology laboratory protocols, containing instructions in both free text and protocol-specific pseudocode. The dataset has been reviewed by domain experts and allows evaluation of model performance several different tasks,

such as next-step prediction, or full protocol generation. We further show the utility of this dataset by automatically designing and successfully executing a lab experiment using GPT-4 and the action space defined using BIOPROT.

In summary, we make the following contributions: (i) We propose evaluating protocol generation on pseudocode rather than free text instructions; (ii) We introduce the BIOPROT dataset, a manually audited dataset of open-access biology protocols; (iii) We evaluate the ability of GPT-4 to accurately convert natural language protocols into pseudocode; (iv) We define a suite of tasks and metrics for evaluating protocol generation; (v) We evaluate several LLMs on our tasks to provide objective measures of these models' ability to generate biological experiments; (vi) We automatically generate a biology experiment and successfully execute it in a lab.

## 2 Related Works

**LLMs for Natural Sciences** Using LLM for scientific tasks such as entity extraction in biological documents (Tamari et al., 2021) or retrieval of chemical reaction procedures (Bai et al., 2022) is a natural use case of such models. Work such as SciBERT, BioGPT, Galactica and others have also shown the utility of pretraining an LLM on a corpus of biomedical (Gu et al., 2021; Lewis et al., 2020; Luo et al., 2022; Lee et al., 2020; Shin et al., 2020) or general scientific text (Beltagy et al., 2019; Taylor et al., 2022). More recently, pre-trained generalist LLMs such as GPT-3 (Brown et al., 2020) and GPT-4 (OpenAI, 2023) have shown to be capable of tasks such as searching for chemical compounds similar to a given one (OpenAI, 2023) or drug editing (Liu et al., 2023c). Furthermore, a GPT-4 agent augmented with tools has been shown to be capable of synthesis planning and drug discovery (Bran et al., 2023) or planning reactions and executing them on a robotic platform (Boiko et al., 2023).

**Task Decomposition** LLMs trained on next-token prediction can struggle with more complex logical reasoning in naïve setups (Liu et al., 2023b). However, decomposing complex tasks into subtasks in frameworks such as Chain-of-Thought reasoning (Wei et al., 2022; Zhang et al., 2023), and its variants such as Least-to-Most (Zhou et al., 2022) and Tree of Thought reasoning (Yao et al., 2023) improves performance in multi-step reasoning problems. In addition to test-time improve-

ments, LLMs also improve in performance when trained on step-by-step reasoning data generated by larger LLMs (Mukherjee et al., 2023; Mu et al., 2023). Task decomposition has also been combined with self-verification through deductive reasoning to improve step-by-step reasoning accuracy (Ling et al., 2023). Here, we approach task decomposition from another angle - we first ask the model to define a discrete set of actions needed to complete a task, and then how to compose them.

**Planning**  Closely related to task decomposition is planning. LLMs have been successful at planning in simulated and real embodied space, both through the use of restricted action space (Ahn et al., 2022; Driess et al., 2023), function/tool search (Wang et al., 2023a; Schick et al., 2023; Shen et al., 2023; Bran et al., 2023; Boiko et al., 2023) and translation of plans into admissible action space (Huang et al., 2022). Planning models have been explicitly combined with Chain-of-Thought reasoning for performance improvement (Mu et al., 2023; Shen et al., 2023). LLM planners can also learn to create their own training curriculum and refine their function use (Wang et al., 2023a). LLM-based planning and reasoning can benefit from writing problems in a machine-readable language such as Planning Domain Definition Language (PDDL) and symbolic logic (Pan et al., 2023; Silver et al., 2023). Furthermore, interactions with simulators and debuggers can be used to improve both plans (Liu et al., 2023a) and value functions that determine the appropriateness of action calls (Ahn et al., 2022; Driess et al., 2023; Mu et al., 2023). Our work extends recent work in planning through the automated generation of admissible action spaces and consequent evaluation without the need for a simulation environment.

**Evaluating LLM Scientists**  Evaluating LLMs on scientific tasks is limited to QA benchmarks for measuring general science knowledge (Hendrycks et al., 2020), or specialist knowledge such as chemistry (Guo et al., 2023; Wu et al., 2017), biomedical science (Sung et al., 2021) or medicine (Jin et al., 2019, 2021). However, evaluating an LLM's performance on more open-ended tasks, such as healthcare support (Dash et al., 2023) or chemical synthesis planning (Bran et al., 2023) is done manually. To the best of our knowledge, we are the first to approach automatic evaluation of LLMs on open-ended problems in science.

**Automatic Evaluation of LLMs**  While evaluation of the performance of an LLM in games (Wang et al., 2023a) or planning in PDDL domains (Silver et al., 2023) can be done automatically, many works rely on self-evaluation, where GPT-4 is used as an evaluator (Bubeck et al., 2023; Bran et al., 2023; Chiang et al., 2023; Peng et al., 2023; Zhou et al., 2023). However, these have been found to contradict human evaluation (Bran et al., 2023) or be systematically biased (Wang et al., 2023b), where the order of the provided responses affects the predicted ranking. In comparison to these works, we use an LLM to generate pseudo-ground truth data on an easy task, in which the model consistently performs well at, which we use to evaluate on a more difficult task with real-world implications.

## 3  The BIOPROT dataset

Here we describe the BIOPROT dataset - a collection of publicly available protocols that are used to evaluate the performance of LLMs on protocol generation on a large range of topics in biology. We discuss the contents of the dataset (Section 3.1), creating a set of admissible actions and translating the protocol steps (Section 3.2), manual verification of the data (Section 3.3), and the tasks that can be approached with it (Section 4). The dataset can be found in the Supplementary Materials. This approach can be used to generate pseudocode datasets in any domain that has step-by-step instructional data.

### 3.1  A Dataset of Protocols for Biology

We collect publicly available protocols from Protocols.io (Teytelman et al., 2016), a platform for developing and sharing reproducible methods. This database contains over 9,000 public protocols of different scientific areas and complexity. Each protocol consists of (i) a title, (ii) a description, and (iii) step-by-step instructions. We automatically and manually filter the protocols, in order to obtain a set of protocols that are related to biology, can be reproduced, and are of sufficient difficulty. For further details about the filtering, refer to the Supplementary Materials. In Table 1 we present a summary of the collected protocols.

### 3.2  Translating Protocols to Pseudocode

As discussed in Section 1, evaluation of planning problems is difficult in natural text, and prior works opt for manual evaluation (Bran et al., 2023; Boiko

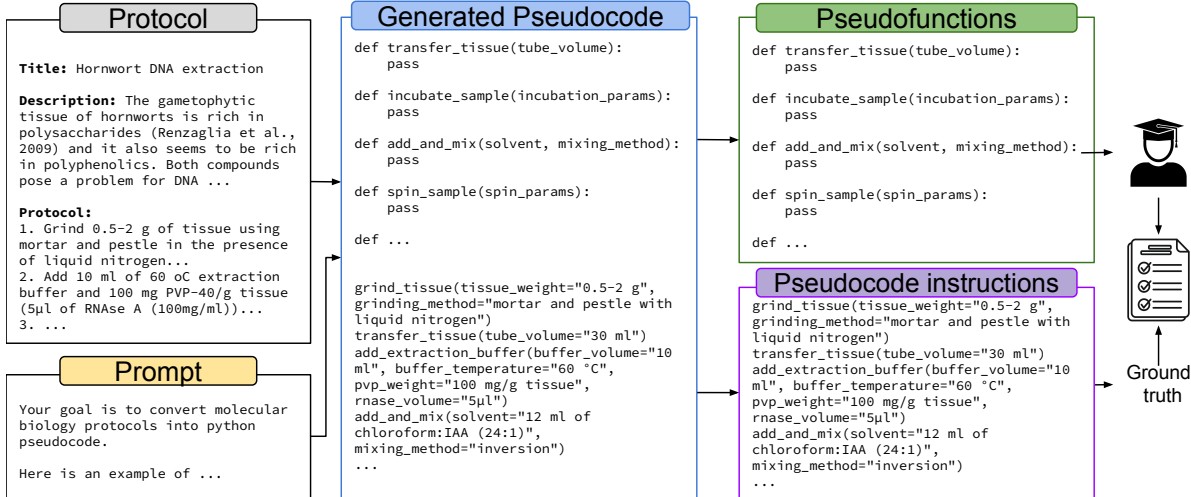

Figure 2: **Creation of pseudofunction and pseudocode data** The model is prompted to generate pseudofunctions and pseudocode based on a target protocol. This generated code is automatically debugged using a feedback error loop, and then manually reviewed. Generated pseudofunctions are used to define the admissible action space in downstream evaluation tasks, and pseudocode instructions using the pseudofunction calls are used as ground truth to measure the accuracy of generated code in downstream tasks, enabling automatic evaluation.

| Statistic | Value |
|---|---|
| Protocols | 100 |
| Average number of steps | 12.5 |
| Average total protocol length in tokens | 641.0 |
| Average tokens per step | 52.6 |
| Average tokens per original description | 83.8 |
| Average tokens per generated description | 66.3 |

Table 1: **Dataset Statistics** We present aggregate statistics for the BIOPROT dataset. The "generated descriptions" are generated using GPT-4 from the step-by-step instructions, as discussed in Section 5.4.

et al., 2023). To this end, we "translate" the free text protocols into pseudocode using GPT-4 (see Figure 2). We task GPT-4 to (i) define a set of *pseudofunctions* that suffice to execute the protocol, and (ii) convert the protocol steps into *pseudocode* using only the provided pseudofunctions.

We make use of a one-shot example prompt, and an automatic feedback loop (Liu et al., 2023a) that provides error signals if: the generated code is not valid Python pseudocode; no pseudofunctions are defined; the pseudocode or pseudofunctions do not have arguments; any numerical parameters in the pseudocode do not have units. Finally, GPT-4 is prompted to check for errors or omissions in the pseudofunctions and pseudocode. Information about our generated pseudocode is summarized in Table 2.

| Statistic | Value |
|---|---|
| Avg. number of pseudofunctions per protocol | 10.3 |
| Avg. number of pseudofunctions per step | 0.82 |
| Avg. number of lines of pseudocode | 17.2 |

Table 2: **Pseudocode Statistics** We present aggregate statistics about the automatically generated pseudofunctions and pseudocode.

### 3.3 Manual Verification

We manually reviewed the generated pseudofunctions and pseudocode for accuracy. Original protocols and generated ground-truth pseudocode were assessed line-by-line by a competent laboratory scientist. They confirmed (i) whether the original natural language protocol made sense, (ii) whether the title and description sufficiently described the protocol so that a competent scientist could attempt to complete it without the protocol, and (iii) whether the pseudocode was accurate. Finally, edits were made to the generated pseudocode as necessary. We show a breakdown of the edits made in Table 3.

| Statistic | Value |
|---|---|
| % generated protocols requiring no edits | 59 |
| % generated protocols with $1 \leq 3$ edited lines | 24 |
| % generated protocols with > 3 edited lines | 17 |
| average number of line edits in edited files | 11.8 |

Table 3: **Manual Verification** We provide a breakdown of the protocols that required manual edits.

Overall, 59 of the 100 protocols were found to be completely accurate requiring no edits. However, many protocols that did require edits only required minor edits. The most common errors found were missing units for numbers, which in most cases would not prevent a competent scientist from completing a protocol. The more impactful errors found were most commonly (1) missing details which would allow one to successfully complete a step of the protocol (these were usually highly verbose steps which explained a detailed technical method for manipulating a sample) and (2) not explaining the composition of a material used in the protocol (e.g. a buffer).

The corrected protocols are made available as the BIOPROT dataset. Even without human editing, LLMs with error-checking loops can be used to create a largely accurate dataset for biology protocol pseudocode, thus enabling self-evaluation.

### 3.4 Machine-generated Descriptions

For some of our downstream tasks, it is necessary to have high-quality descriptions of protocols that give a sense of what the protocol steps should include. However, protocol descriptions in Protocols.io are not always suitable for this purpose. To this end, we also generated descriptions of protocols that provided a high-level overview of the protocols' objective (the prompt for this is seen in the Supplementary Materials). We include both our machine-generated descriptions and the original descriptions in our dataset.

## 4 Metrics and evaluation

Using the BIOPROT dataset, we evaluate an LLM's capabilities to reason about and generate scientific protocols on several different tasks.

**Next Step Prediction**   Given a protocol title, description, an admissible set of pseudofunctions, and partially completed pseudocode, we evaluate the model's ability to correctly identify the pseudofunction corresponding to the next step in the protocol. We evaluate the correctness of both the predicted function and the function arguments.

For function-level accuracy, we report the percentage of the number of correct function assignments

$$\text{accuracy} = \frac{1}{N} \sum_{n=1}^{N} \mathbb{1}[f_i^{\text{pred}} = f_i^{GT}],$$

where $f^{\text{pred}}$ and $f^{GT}$ are the predicted and groundtruth functions, respectively, and $N$ is the number of steps in the protocol.

During generation, the model is prompted to name each function argument and provide the argument parameters. To evaluate accuracy of the arguments, we first check whether the function argument names is correct. For that purpose, we compute precision and recall of the arguments' names. For correct function arguments, we consider the accuracy of the argument value using the BLEU metric (Papineni et al., 2002). Additionally, we encode the predicted and ground truth argument values, $a_i^{\text{pred}}$ and $a_i^{\text{GT}}$, respectively, with SciBERT (Beltagy et al., 2019) sentence encoder $\mathcal{E}$ to get the SciBERTscore:

$$\text{SciBERTscore} = \frac{1}{N} \sum_{i=0}^{N} \frac{\langle \mathcal{E}(a_i^{\text{pred}}), \mathcal{E}(a_i^{\text{GT}}) \rangle}{\|\mathcal{E}(a_i^{\text{pred}})\| \|\mathcal{E}(a_i^{\text{GT}})\|},$$

which is the average cosine similarity between predicted and ground truth argument values for all $N$ steps. This metric is inspired by BERTScore (Zhang et al., 2019), but we use a SciBERT encoder as it is better suited to the scientific domain. We only compute argument-level metrics for correctly predicted functions, as not to penalize the model twice for wrong function predictions.

**Protocol Generation**   Given a protocol title, description, and an admissible set of pseudofunctions, the model is tasked to generate corresponding pseudocode. We again evaluate the correctness of predicted functions and their corresponding arguments. This is a more difficult task than the previous one, as the model needs to plan the entire execution of the protocol. For function-level evaluation, we need to measure (i) if the correct functions were called, and (ii) if they were used in the correct order. For the former, we report precision and recall of function calls, where we take into account repeated calls of the same function. For evaluating whether the functions are used in the correct order, we use the Levenshtein distance $\mathcal{L}_d$ between the predicted and ground-truth sequence of functions. The Levenshtein distance is originally a string edit distance that measures the number of insertions, deletions, or substitutions to make one word into another. We consider each function call as a separate symbol, which incurs a cost of 1 for being added, deleted, or substituted. We report a normalized Levenshtein

| Model | Shuffle | Functions Accuracy | Arguments | | | |
|---|---|---|---|---|---|---|
| | | | Precision | Recall | SciBERTScore | BLEU |
| GPT-3.5 | ✗ | 65. $\pm$ 1.3 | 97.7 $\pm$ 0.5 | 94.7 $\pm$ 0.5 | 88.5 $\pm$ 0.5 | 0.363 $\pm$ 0.012 |
| GPT-3.5 | ✓ | 36.1 $\pm$ 1.6 | **97.1** $\pm$ 1.2 | **95.1** $\pm$ 1.0 | **88.6** $\pm$ 0.5 | **0.384** $\pm$ 0.028 |
| GPT-4 | ✗ | **70.6** $\pm$ 0.4 | **97.1** $\pm$ 0.5 | 94.9 $\pm$ 0.6 | 87.9 $\pm$ 0.5 | 0.351 $\pm$ 0.017 |
| GPT-4 | ✓ | 57.0 $\pm$ 0.8 | **97.1** $\pm$ 0.4 | 94.7 $\pm$ 0.8 | 88.5 $\pm$ 0.6 | 0.363 $\pm$ 0.025 |

Table 4: **Next Step Prediction Evaluation** Given a protocol title and description, the admissible pseudofunctions and partially completed pseudocode, we evaluate the model's ability to correctly predict the next step. For all metrics, higher is better. We report mean and standard deviation over 5 runs.

distance $\mathcal{L}_{dn}$

$$\mathcal{L}_{dn} = \frac{\mathcal{L}_d}{N},$$

where $N$ is the number of functions in the ground-truth pseudocode.

In addition, we evaluate the predicted function arguments. We use the same metrics as described under "Next Step Prediction".

**Function Retrieval** Our approach has the potential to allow novel protocols to be assembled from steps provided in existing protocols in the dataset, if the model is able to correctly identify which steps are needed for any given protocol. Thus, given a protocol title and description, and a set of pseudofunctions, we evaluate the models' ability to correctly identify which of the provided functions are needed to execute the protocol. In this task, we provide the model with a set of pseudofunctions consisting of the ground-truth pseudofunctions for the given protocol, and pseudofunctions drawn from several (i) random or (ii) nearest neighbour protocols. Providing functions from nearest neighbour protocols is more difficult, as they are likely to be more similar to the correct functions. We measure the precision and recall of retrieved functions.

## 5 Experiments

### 5.1 Implementation details

We explore the performance of GPT-3.5 and GPT-4 from the OpenAI API. Where we find nearest neighbors, we use an embedding index of all protocols' descriptions using `text-embedding-ada-002` embeddings, unless stated otherwise. We show the prompts we use in the Supplementary Material.

For each of the tasks listed in Section 4, we evaluate the models in several settings:

- **Shuffled**: the model can be provided either with functions in the order in which they are generated, or randomly shuffled. The functions tend to be defined in the order they appear in the original protocol, and that serves as a signal to the model we evaluate. By randomly shuffling the input functions, we make the task more difficult.

- **Feedback**: The model has access to an error loop that can detect undefined functions and Python syntax errors. Such feedback loops have been found to be beneficial in PDDL planning (Silver et al., 2023) and reasoning (Madaan et al., 2023).

### 5.2 Results

**Next step prediction** We show results on next step prediction in Table 4. We see that GPT-4 consistently outperforms GPT-3.5 in both the prediction of the correct next step, whereas GPT-3.5 performs better at predicting function arguments. We note there is a drop in performance when the input functions are shuffled, likely because if not shuffled, the functions appear roughly in the order as they should be called as they were sequentially generated by the LLM.

**Protocol generation** We show results on full protocol generation in Table 5. We observe the biggest gap in the Levenshtein distance score metric, where GPT-4 significantly outperforms GPT-3.5. Meanwhile, GPT-4 and GPT-3.5 show similar precision and recall of used functions. This suggests that while both have a similar ability to use the correct functions, GPT-4 performs better at using the right order. We also observe that shuffling the input functions consistently leads to a drop in performance.

**Function retrieval** We show retrieval results in Table 6. We see that GPT-4 outperforms GPT-3.5 on this task. However, the results on this task appear generally poor. One possible reason for the

| Model | Shuffle | Feedback | Functions | | | Arguments | | | |
|-------|---------|----------|-----------|--------|-----------------------|-----------|--------|--------------|--------|
| | | | Precision | Recall | $\mathcal{L}_{dn} \downarrow$ | Precision | Recall | SciBERTScore | BLEU |
| GPT-3.5 | ✗ | ✗ | **93.4** ± 0.9 | 89.9 ± 0.6 | 0.498 ± 0.036 | 72.7 ± 0.8 | 91.4 ± 1.5 | 82.7 ± 0.6 | 0.121 ± 0.005 |
| GPT-3.5 | ✗ | ✓ | 93.3 ± 1.0 | **91.1** ± 1.1 | 0.505 ± 0.159 | 73.1 ± 1.6 | 88.1 ± 1.9 | **82.8** ± 0.6 | 0.117 ± 0.006 |
| GPT-3.5 | ✓ | ✗ | 91.8 ± 0.8 | 85.9 ± 2.8 | 0.945 ± 0.055 | 72.9 ± 1.4 | 89.1 ± 2.2 | 81.8 ± 0.2 | 0.102 ± 0.003 |
| GPT-3.5 | ✓ | ✓ | 92.5 ± 0.3 | 86.1 ± 1.6 | 0.884 ± 0.045 | 73.2 ± 1.3 | 87.3 ± 3.5 | 82.3 ± 0.4 | 0.102 ± 0.009 |
| GPT-4 | ✗ | ✗ | 91.9 ± 0.9 | 90.8 ± 0.9 | **0.396** ± 0.046 | 72.2 ± 0.8 | **94.7** ± 1.4 | 82.6 ± 0.2 | **0.124** ± 0.006 |
| GPT-4 | ✗ | ✓ | 92.5 ± 0.3 | 90.1 ± 0.3 | 0.438 ± 0.412 | 72.0 ± 0.3 | 93.3 ± 1.0 | 82.7 ± 0.3 | 0.112 ± 0.005 |
| GPT-4 | ✓ | ✗ | 92.6 ± 0.9 | 87.7 ± 0.9 | 0.722 ± 0.311 | 72.2 ± 0.3 | 94.6 ± 1.8 | 82.7 ± 0.4 | 0.113 ± 0.004 |
| GPT-4 | ✓ | ✓ | 92.8 ± 1.0 | 86.6 ± 0.3 | 0.685 ± 0.178 | **73.7** ± 0.7 | 93.4 ± 2.0 | 82.5 ± 0.7 | 0.108 ± 0.004 |

Table 5: **Protocol Generation Evaluation** Given a protocol title and description, and a set of admissible pseudo-functions, we evaluate the model performance on full protocol generation. For all metrics higher values are better, except for the normalized Levenshtein distance $\mathcal{L}_d n$, where lower values are better. Best performance is bolded and second best is underlined. We report mean and standard deviation over 5 runs.

| Model | Neighbourhood | Precision | Recall |
|-------|---------------|-----------|--------|
| GPT-3.5 | Nearest | 24.2 | 35.7 |
| GPT-3.5 | Random | 36.7 | 45.2 |
| GPT-4 | Nearest | 32.5 | 39.2 |
| GPT-4 | Random | 48.8 | 49.4 |

Table 6: **Function retrieval.** Performance on function retrieval of pseudofunctions from the query protocol, as well as (i) random or (ii) nearest neighbors protocols.

poor performance is that the correct answer may sometimes be ambiguous. For example, `Mix` and `MixSubstance` are semantically identical, but have different syntax, and the model would be penalized for selecting a function not from the query protocol. This effect would explain why performance using the""nearest" neighbours is worse than performance when using "random" protocols.

### 5.3 Using GPT-4 as an evaluator

We use GPT-4 as an evaluator, where given (i) a protocol description, (ii) admissible pseudofunctions, (iii) ground-truth pseudocode (generated as described in Section 3.2), and (iv) predicted pseudocode, the model is prompted to predict which one of (iii) or (iv) better matches the protocol description (i). We report the rate at which the predicted pseudocode was preferred in Table 8. In general, GPT-4 only performs slightly above chance in identifying the ground truth protocol, versus LLM generations, although it is unclear whether this is because the machine-generated protocols are largely correct, or because GPT-4 is unable to distinguish correct from incorrect protocols. Note that prior works (Bran et al., 2023) found that a GPT evaluator tends to prefer longer and more coherent, but not necessarily more correct generations.

### 5.4 Using GPT-4-Generated Descriptions

For some protocols, we observe that the detail present in the protocol description does not suffice to enable protocol reconstruction. To this end, we use GPT-4 to generate a short pseudo description given the protocol steps in natural text. We present results on next step generation and full protocol generation in Figure 8. We see a small increase in performance, which is expected, as the summary-generating model can include more detail (however, the pseudo descriptions are shorter – see Table 2).

### 5.5 Real-World Validation

Finally, to validate that BIOPROT can be used to generate accurate novel protocols, we devised a setup for end-to-end protocol creation. To do this we opted to build an LLM agent with access to tools, such that it can retrieve protocols that contain relevant pseudofunctions, and use their pseudofunctions to generate new pseudocode. Note that for good performance in this real-world validation task, the LLM needs to be able to (1) find relevant psueodofunctions from other protocols, and (2) generate correct pseudocode, both of which are tasks we build metrics for. Details are as follows: we created a Toolformer-like (Schick et al., 2023) chain-of-thought LLM agent (Wei et al., 2022) with access to a tool for searching for protocols in the BIOPROT database. This agent used the GPT-4 LLM. We prompted the agent to retrieve protocols relevant to generating a new target protocol. We extracted the pseudofunctions from the retrieved protocols and then prompted the agent to generate a new protocol using only the retrieved pseudofunctions. We used this setup to create two experiments using GPT-4: (1) culturing a single colony of E.coli bacteria overnight and making a glycerol stock with

| Model | Description generated by | Accuracy | Functions Precision | Recall | $\mathcal{L}_{dn} \downarrow$ | Arguments Precision | Recall | SciBERTScore | BLEU |
|---|---|---|---|---|---|---|---|---|---|
| GPT-4 | 🧑‍🔬 | 46.1 | – | – | – | 98.1 | **95.6** | 88.9 | 0.334 |
| GPT-4 | Ⓢ | **48.4** | – | – | – | **98.4** | 95.1 | **90.0** | **0.393** |
| GPT-4 | 🧑‍🔬 | – | 91.1 | 90.1 | 0.49 | 71.8 | 95.5 | 84.1 | **0.126** |
| GPT-4 | Ⓢ | – | **92.2** | **90.3** | **0.45** | **73.3** | **95.8** | **84.5** | 0.122 |

Table 7: **Using GPT-4 - generated description** We compare performance on next step prediction (top) and protocol generation (bottom) when using a protocol description generated by (i) scientists, or (ii) GPT-4. We see that using a GPT-4 generated description consistently outperforms the original one. The input pseudofunctions to the model are shuffled and we use a feedback loop.

| Model | Shuffle | Feedback | GPT-4 score ↑ |
|---|---|---|---|
| GPT-3.5 | ✗ | ✗ | 35.6 |
| GPT-3.5 | ✗ | ✓ | 40.2 |
| GPT-3.5 | ✓ | ✗ | 40.9 |
| GPT-3.5 | ✓ | ✓ | 39.3 |
| GPT-4 | ✗ | ✗ | 43.9 |
| GPT-4 | ✗ | ✓ | 42.4 |
| GPT-4 | ✓ | ✗ | 40.9 |
| GPT-4 | ✓ | ✓ | 42.4 |

Table 8: **GPT-4 as an evaluator.** The GPT-4 score shows the rate at which GPT-4 predicted the model's output to be better than the ground truth.

the suspension (a form of cryopreservation for long-term storage), and (2) culturing Symbiodinum (a large genus of dinoflagellates endosymbiontic to cnidarians that may help corals survive in warming oceans), extracting its DNA, and then running the DNA on an agarose gel.

## 5.6 Real-World Validation Results

The model generated two new protocols using pseudofunctions from our database. Both of these protocols were reviewed by a scientist and were determined to be accurate and sufficient for a competent lab scientist to follow. We opted to complete the first protocol using E.coli as we did not have Symbiodinium available in the laboratory. We validated the first protocol by implementing it in the lab with the instructions and parameter values provided by the model. The protocol ran successfully: the cells remained viable after storage at -80 °C, as evidenced by subsequent culture on nutrient agar (see Figure 3). The methods and prompts used to generate these experiments, as well as the agent chain-of-thought reasoning, can be found in the Appendix.

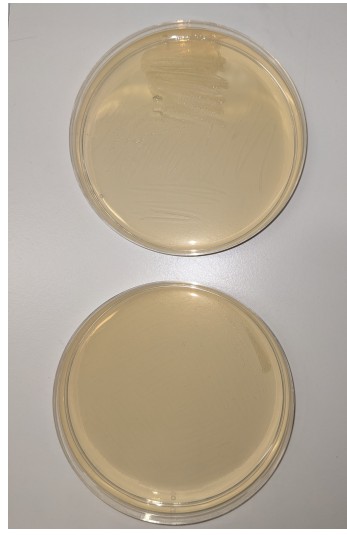

Figure 3: **E.coli growing on nutrient agar plates.** We carried out a protocol for overnight culture and cryopreservation of E.coli in glycerol for long-term storage. One hour after completion of the protocol, the cells were thawed and spread onto the surface of nutrient agar. After 10 hours they can be seen growing on the surface of the agar plate (top) plate, while there is no growth on the control (no E.coli) plate (bottom). This shows the LLM-generated protocol was correct.

## 6 Conclusion

We have introduced a method for automatic evaluation of LLMs on open-ended planning problems, such as those found in experimental sciences, and a dataset of such planning problems in biology laboratory protocols. We then defined a suite of tasks and evaluation metrics that can be used to measure performance and help drive progress in the field. We evaluate GPT-3.5 and GPT-4 on these tasks and find that there is more to be desired in terms of performance. Finally, we show an application of our dataset and framework, where an LLM generates a protocol that is successfully executed in a laboratory.

# 7 Limitations

**Use of paid API**    The GPT-4 and GPT-3.5 models we use are not open-sourced and can have significant costs for large-scale experiments. In total, we used approximately $1000 for API calls. Further work should explore the performance of open-sourced LLMs.

**Additional scientific fields**    Our work is focused on biology, but could be extended to other fields such as chemistry and materials science. Future works should explore extending the dataset and framework.

**Misuse**    There is a risk of misuse, where adversaries could use our framework or dataset to inform the synthesis of harmful compounds. We have taken care to ensure the protocols in BIOPROT contain no protocols that can easily be used for such purposes. Continued research on aligning LLMs and restriction of dangerous outputs is important to minimize risk. We hope that our approach of using pseudofunctions may in the future allow for easier programmatic evaluation of outputs, and easier detection of the generation of hazardous substances.

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

# BioPlanner: Automatic Evaluation of LLMs on Protocol Planning in Biology

**Appendix**

## A Dataset Filtering

We filter Protocols.io protocols such that (i) they can be parsed and provided to an LLM, and (ii) they are sufficiently challenging to serve as an evaluation set. We performed the following automatic and manual filtering:

**Automatic filtering** Protocols were automatically removed if they:

- Do not contain a description
- Contain linked files that could not be parsed
- Contain images that could not be parsed by text-only models
- Contain tables (standard LLMs can sometimes struggle with table-based data representations without few-shot examples (Hegselmann et al., 2023))
- Consist of fewer than three steps (such protocols were insufficiently complex to demonstrate multi-step planning)

**Manual filtering** Following automatic filtering, protocols were manually removed if they:

- Were not relevant to biology
- Were considered poorly written to the extent that a human could not accurately replicate the protocol

## B Prompts

### B.1 Main experiments

We show the prompts we use for generating pseudofunctions and pseudocode in Figure 6, predicting pseudocode in Figure 7, summarising protocol steps in Figure 8. We show the error messages we use in the feedback loops in Figure 9. Figure 12, and the resulting generated protocol used in our real-world experiment is found in Figure 13

### B.2 Lab experiments

Here we show the prompts we use in Section 5 of the paper. Figure 10 is the prompt provided to the CoT agent. Figure 11 shows the Langchain output form the agent. Figure 12 shows the prompt that contains the retrieved pseudofunctions. Finally, Figure 13 shows the pseudocode that was given to a biologist to execute in a laboratory.

## C Qualitative Evaluation

We show qualitative results for protocol id 145 from BIOPROT. For further qualitative examples, please refer to the BIOPROT dataset.

**Title** Ethanol precipitation of nucleic acids (Eppendorf tubes)

**Description** Nucleic acid precipitation is used to concentrate and/or purify nucleic acids. The below protocol is based on the fact that nucleic acids are less soluble in alcohol than in more polar water. Addition of salt further decreases solubility by competing for water dipoles; as does low temperature. Please see the OpenWetWare website for more details.

**Steps**

1. Add 1/10 volume of 3M sodium acetate, pH 5.2 or 1/2 volume of 5M ammonium acetate. reagents

2. Add 2-3 volumes of 100% Ethanol.

3. Mix and freeze overnight in -20. NOTES In general, the time you need to incubate in the freezer depends on how much nucleic acid you have, how big it is and the volume it is in. My general protocol is to freeze for 20 min to 1 hr at -80C. This seems to work well for most things, but you may want to freeze longer if you have only a small concentration of nucleic acid or if it is small in size(<15 nucleotides). (Kathleen) NOTES If you are in a hurry, you can also dip you epi shortly into liquid nitrogen. If you added enough ethanol, the mix won't freeze. Careful with isopropanol - it freezes more quickly. This works well for me and saves me a lengthy incubation in the fridge. (Jasu)

4. Spin at full speed in a standard microcentrifuge at 4 degrees for 30 minutes. 1800s

5. Decant (or carefully pipet off) the supernatant.

6. Dry the pellet. NOTES For this you can air dry (tubes open, 15 min) or dry in a speedvac. DNA and RNA (if you don't have RNases in your sample) are typically hearty

| Model | Shuffle | Feedback | Functions | | | Arguments | | | |
|---|---|---|---|---|---|---|---|---|---|
| | | | Precision | Recall | $\mathcal{L}_{dn}\downarrow$ | Precision | Recall | SciBERTScore | BLEU |
| Llama2-7B | ✗ | ✗ | 83.6 | 49.8 | 0.74 | 76.2 | 41.4 | 79.8 | 0.048 |
| Llama2-7B | ✗ | ✓ | 81.0 | 45.9 | 0.82 | 70.4 | 42.9 | 80.4 | 0.050 |
| Llama2-7B | ✓ | ✗ | 82.2 | 45.1 | 0.63 | 70.7 | 43.8 | 81.3 | 0.051 |
| Llama2-7B | ✓ | ✓ | 78.5 | 30.4 | 0.56 | 73.0 | 51.4 | 81.1 | 0.047 |

Table 9: **Evaluating Llama on Protocol Generation Evaluation** Given a protocol title and description, and a set of admissible pseudofunctions, we evaluate the model performance on full protocol generation. For all metrics higher values are better, except for the normalized Levenshtein distance $\mathcal{L}_d n$, where lower values are better.

| Model | Neighbourhood | Precision | Recall |
|---|---|---|---|
| LLama2-7B | Nearest | 26.1 | 57.5 |
| LLama2-7B | Random | 28.1 | 56.3 |

Table 10: **Evaluating Llama on Function retrieval.** Performance on function retrieval of pseudofunctions from the query protocol, as well as (i) random or (ii) nearest neighbors protocols.

enough for you to air dry at 37C, if desired. NOTES Overdrying can make DNA hard to re-dissolve. Especially for longer DNA, I avoid vacuum drying and airdry only briefly before re-dissolving. (Jasu)

7. Add your desired quantity of water. Vortex and spin down to resuspend. NOTES Beware of using water unless you are sure of what you are getting in to. The "pH" of water can vary widely (I've seen from pH 5 to pH 8.5), and depurination of DNA at low pH or degradation of RNA at high pH are possibilities. Water also typically contains trace metals, which can accelerate these reactions. I typically recommend resuspension in TE (10 mM Tris-HCl, pH 7.5, 1 mM EDTA). This makes sure your nucleic acid is at a neutral pH and the EDTA will chelate any trace metals. Since they are in such small amounts, neither the buffer nor the EDTA will affect most downstream reactions. (Kathleen)

**Generated Pseudocode and Pseudofunctions** We show the generated pseudocode and pseudo-functions, which we use as ground truth, in Figure 4

**Predicted Protocol** We show the predicted protocol in Figure 5

## D LLama evaluation

To benchmark performance on open-source models, we also conducted a run of our experimental evaluation tasks on Llama-2 (Touvron et al., 2023). We evaluate the 7B model and report performance on protocol generation and function retrieval in Table 9 and Table 10, respectively. We found that Llama-2 significantly underperforms GPT-3.5 and GPT-4 models in function selection. As part of our evaluation on Llama-2 we observe that, when using feedback, the model is distracted and does not attempt to re-write code. Iterative feedback appears to be a process that is effective for GPT models and not Llama models, and this observation is consistent with prior work (Madaan et al., 2023). We also ran Llama-2 on the next step prediction task, but we found that the model was unable to complete this task. The model would typically produce text that states an intent to complete the pseudocode rather than writing the actual next pseudocode line. This difference in behaviour is likely due to a difference in training regimes between GPT models and Lllama-2, but given the lack of documentation around the training of GPT models, the precise nature of this difference is unknown.

## E Dataset and Evaluation

The BIOPROT dataset and evaluation metrics from this paper can be found at https://github.com/bioplanner/bioplanner

## F Human Benchmarking

While we believe our metric is internally useful for comparing the performance of LLM models and approaches, we wanted to assess how our tasks used relate to human performance. To this end, we performed a human evaluation of next-step prediction tasks and function selection tasks. We worked with an undergraduate biomedical sciences student and asked them to complete the next step prediction task and the function selection task. The student had access to internet search and an unlimited amount of time to answer questions. With the next step prediction task we provided shuffled

functions, and with the function selection task, we used random distractor functions. For the function selection task, Human Precision was 87.5%, and human Recall was 0.84%, (n=20), indicating a significant increase in performance over GPT-4. GPT-4 performance is potentially weaker than human performance in the function selection task due to the large number of nearest-neighbour functions in the context window acting as distractors from the task instructions. For the next step prediction task human accuracy was 54.8% (n=32), with Precision and Recall or arguments being 97% and 95% respectively. This performance is roughly comparable to GPT-4 in the shuffled function setting.

```
# Functions
def add_salt(solution, salt, volume_ratio):
    pass

def add_ethanol(solution, volumes_ethanol):
    pass

def mix_solution(solution):
    pass

def incubate_freeze(solution, temperature, time):
    pass

def spin_tube(solution, spin_params):
    pass

def remove_supernatant(solution):
    pass

def dry_pellet(pellet, method, time):
    pass

def resuspend_pellet(pellet, solvent, volume):
    pass

# Protocol steps
add_salt(solution="nucleic_acid_solution", salt="3M sodium acetate, pH 5.2", volume_ratio=1/10)
add_ethanol(solution="nucleic_acid_solution", volumes_ethanol=2)
mix_solution(solution="nucleic_acid_solution")
incubate_freeze(solution="nucleic_acid_solution", temperature=-20, time="overnight")
spin_tube(solution="nucleic_acid_solution", spin_params="full speed, 4 degrees, 30 minutes")
remove_supernatant(solution="nucleic_acid_solution")
dry_pellet(pellet="pellet", method="air_dry", time="~15 min")
resuspend_pellet(pellet="pellet_a", solvent="TE_buffer", volume="desired volume")
# end of protocol
```

Figure 4: **Generated pseudofunctions and pseudocode.** Given the protocol title, description, and free text step-by-step instructions, we generate pseudocode and pseudofunctions.

```
# raw protocol
add_salt(solution="nucleic_acid_solution", salt="0.3 M Sodium Acetate or 0.2 M NaCl", volume_ratio="1/10 of the solution volume")
add_ethanol(solution="nucleic_acid_solution", volumes_ethanol=2.5)
mix_solution(solution="nucleic_acid_solution")
incubate_freeze(solution="nucleic_acid_solution", temperature="-20°C", time="30 minutes to overnight")
spin_tube(solution="nucleic_acid_solution", spin_params="12,000 x g, 4°C, 20 minutes")
remove_supernatant(solution="nucleic_acid_solution")
add_ethanol(solution="pellet", volumes_ethanol=1)
mix_solution(solution="pellet")
spin_tube(solution="pellet", spin_params="12,000 x g, 4°C, 5 minutes")
remove_supernatant(solution="pellet")
dry_pellet(pellet="pellet", method="air dry or vacuum centrifuge", time="approximately 5 minutes")
resuspend_pellet(pellet="pellet", solvent="water or TE buffer", volume="appropriate for downstream applications")
```

Figure 5: **Predicted pseudocode.** Given protocol title, description, and an admissible set of pseudofunctions, a model predicts pseudocode. This coresponds to the pseudofunctions in Figure 4.

```
pseudo_groundtruth_prompt = """
Your goal is to convert molecular biology protocols into python pseudocode.

EXAMPLE
Here is an example of how to convert a protocol for {example_description}

example protocol:

{example protocol steps}

example python pseudocode:

{example_pseudofunctions}

# Protocol steps

{example_pseudocode}

YOUR TASK:
Here is a molecular biology protocol entitled '{title}' The protocol steps are as follows:

{protocol}

Please convert this protocol into python pseudocode. Please define the python functions you will \
use at the start of the pseudocode, and ensure that these functions have parameters where appropriate.

python pseudocode:
"""
```

Figure 6: **Prompt for generating pseudofunctions and pseudocode.**

```
pseudocode_prompt = """
Your goal is to generate python pseudocode for molecular biology protocols. \
This pseudocode must accurately describe a complete scientific protocol to obtain \
a result, and must only use a pre-defined list of pseudocode functions.

Here is an example of how to generate pseudocode for a molecular biology protocol.

EXAMPLE:

{example title}

Here are some extra details about the protocol:

{example details}

example pseudocode:

{example pseudocode}

YOUR TASK:
Generate pseudocode for a protocol for {title}.

Here are some extra details about the protocol:

{details}

You may only make use of the following python pseudocode functions:

{psuedofunctions}

your pseudocode:"""
```

Figure 7: **Prompt for predicting pseudocode.**

```
summary_prompt = """
Here is a molecular biology protocol entitled '{title}'.

Here is the protocol abstract:

{description}

The protocol steps are as follows:

{protocol_steps}

Please provide a 2 sentence summary of the objective of this scientific protocol."""
```

Figure 8: **Prompt for summarizing a protocol.**

```
error_messages = {
    "IndentationError": "IndentationError: There are indentation errors in one or \
        more of your function definitions. Functions should have at least one indented \
        line that has code and not comments in it. This can simply be a return or pass \
        statement.",
    "SyntaxError": "SyntaxError: There are syntax errors in one or more of your function \
        definitions or in your main code.",
    "NoParametersWarning": "NoParametersWarning: One or more of your functions does not \
        have any parameters. Functions typically have parameters to better describe the \
        experiment.",
    "CommentError": "CommentError: Your function includes a comment or docstring. Functions \
        should be named to sufficiently describe their purpose without the need for comments. \
        Please remove the comment or docstring and try again. Consider renaming your function \
        to better describe its purpose.",
    "MissingUnitsWarning": "MissingUnitsWarning: One or more of your function parameter \
        calls include numbers but do not include the units for said numbers. You may be missing \
        units that are important for describing the experiment.",
    "NoFunctionsError": "NoFunctionsError: There do not appear to be any functions in your \
        code. Please define functions",
    "NewFunctionError": "NewFunctionError: You have defined a new function. Please only use \
        the pseudofunctions provided.",
    "NoMainCodeError": "NoMainCodeError: You have not written any pseudocode in the main \
        code section.",
    "UndefinedFunctionError": "UndefinedFunctionError: You have used a function that is not \
        defined in the pseudofunctions provided.",
    "MissingUnitsWarning": "MissingUnitsWarning: One or more of your function parameter \
        calls include numbers but do not include the units for said numbers. You may be \
        missing units that are important for describing the experiment.",
}
```

Figure 9: **Error messages for feedback loops.**

```
query = """
I would like to make a protocol for culturing a single colony of E.coli overnight and then make \
a glycerol stock with the suspension. Could you find me protocols for both parts of this experiment?
"""
```

Figure 10: **LLM query for protocol retrieval**.

```
> Entering new AgentExecutor chain...
I need to search for protocols for culturing a single colony of E.coli overnight and making a gl
ycerol stock with the suspension.
Action: search_for_protocol
Action Input: culturing a single colony of E.coli overnight
Observation: Here are the titles for the top results:
Overnight Bacterial Batch Culture
High Efficiency Transformation Protocol using NEB 10-beta Competent E. coli (C3019H)
Heat Shock Transformation
Thought:I found a relevant protocol for culturing a single colony of E.coli overnight. Now, I ne
ed to search for a protocol on making a glycerol stock with the suspension.
Action: search_for_protocol
Action Input: making a glycerol stock with E.coli suspension
Observation: Here are the titles for the top results:
Glycerol Stock
Overnight Bacterial Batch Culture
Mounting Media for Immunohistochemistry - Drosophila
Thought:I now know the final answer
Final Answer: I found two relevant protocols for your experiment. For culturing a single colony
of E.coli overnight, use the "Overnight Bacterial Batch Culture" protocol. For making a glycerol
 stock with the suspension, use the "Glycerol Stock" protocol.
```

Figure 11: **Langchain output for protocol retrieval**.

```python
prompt = """
Your job is to write python pseudocode to describe scientific protocols for molecular \
biology. I would like you to write a protocol for culturing a single colony of E.coli \
overnight and then make a glycerol stock with the suspension. Your pseudocode may only \
make use of the following functions:

def create_screw_top_tube():
    pass
def add_to_tube(tube, volume, substance):
    pass
def mix_by_inversion(tube):
    pass
def store_in_freezer(tube, temperature):
    pass
def prepare_media_volume(media, volume, tube):
    pass
def add_antibiotic_to_media(antibiotic, concentration, media):
    pass
def inoculate_culture_with_bacteria(source, media):
    pass
def incubate_culture_at_conditions(temperature, time, shaking):
    pass
"""
```

Figure 12: **Prompt to generate protocol from retrieved functions**.

```python
def culture_and_make_glycerol_stock():
    # Step 1: Prepare media with antibiotics
    screw_top_tube_media = create_screw_top_tube()
    add_to_tube(screw_top_tube_media, 5, "LB")  # 5 mL of LB media
    add_antibiotic_to_media(
        "ampicillin", 100, screw_top_tube_media
    )  # Add ampicillin at 100 ug/mL concentration

    # Step 2: Inoculate culture with a single colony of E.coli
    inoculate_culture_with_bacteria("E.coli_colony", screw_top_tube_media)

    # Step 3: Incubate culture overnight
    incubate_culture_at_conditions(37, "overnight", True)  # 37°C, overnight, shaking

    # Step 4: Prepare glycerol stock
    screw_top_tube_glycerol_stock = create_screw_top_tube()
    add_to_tube(
        screw_top_tube_glycerol_stock, 0.5, "glycerol"
    )  # 0.5 mL of 50% glycerol

    # Step 5: Add 0.5 mL of overnight culture to glycerol stock
    add_to_tube(screw_top_tube_glycerol_stock, 0.5, "overnight_culture")

    # Step 6: Mix the glycerol stock by inversion
    mix_by_inversion(screw_top_tube_glycerol_stock)

    # Step 7: Store the glycerol stock in the -80°C freezer
    store_in_freezer(screw_top_tube_glycerol_stock, -80)

# Execute the protocol
culture_and_make_glycerol_stock()
```

Figure 13: **The LLM generated protocol used in our lab experiment**.