# OpenReview forum: "BioPlanner: Automatic Evaluation of LLMs on Protocol Planning in Biology"
_EMNLP/2023/Conference — EMNLP 2023 Main_

### Official Review · Reviewer_3v5p · 2023-08-04

**Soundness:** 3

**Excitement:**

3: Ambivalent: It has merits (e.g., it reports state-of-the-art results, the idea is nice), but there are key weaknesses (e.g., it describes incremental work), and it can significantly benefit from another round of revision. However, I won't object to accepting it if my co-reviewers champion it.

**Paper Topic And Main Contributions:**

This paper explores the application of LLMs in biology protocol planning, with a primary emphasis on devising an automated evaluation method. To facilitate this research, the authors curated a dataset called BioPort, comprising published biology protocols alongside machine-generated pseudocode, which was verified by human experts. In addition, they propose various quantitative tasks and metrics to assess the performance of LLMs in protocol planning. The evaluation results for GPT-3 and GPT-4 are presented, and the model-generated protocols are validated in a real-world laboratory experiment.

**Reasons To Accept:**

1. This paper explores an interesting application of utilizing LLMs for biology experiments, showcasing the potential of NLP techniques to drive scientific discovery.
2. The authors construct a novel dataset comprising natural language descriptions and step-by-step pseudocode.
3. The introduction of three quantitative tasks allows for automatic evaluation of LLM's performance in protocol planning.
4. The machine-generated protocols have been validated in a real-world laboratory experiment.

**Reasons To Reject:**

1. The writing and organization of the paper could benefit from improvement. Typos and missing table references need to be addressed, and the content should be better structured. While the paper contributes to dataset, evaluation, method, and experimental validation, they appear to be somewhat intertwined.
2. Concerns arise regarding the evaluation conducted on machine-generated data, which may limit the dataset's coverage. As the pseudocode is generated by GPT-4 (though manually verified), the dataset's scope may be bound by GPT-4's capabilities.
3. Evaluating GPT-4 on a dataset generated by GPT-4 may not provide the most meaningful insights and might lead to potential overestimation of the model's performance.
4. The dataset's size is relatively small, consisting of only 70 protocols, which might impact the generalizability and robustness of the findings.

**Reproducibility:**

4: Could mostly reproduce the results, but there may be some variation because of sample variance or minor variations in their interpretation of the protocol or method.

**Reviewer Confidence:**

4: Quite sure. I tried to check the important points carefully. It's unlikely, though conceivable, that I missed something that should affect my ratings.

---

> ### Author Rebuttal · Authors · 2023-08-28
>
> We thank the reviewer for their thoughtful review and useful feedback.
> - **Writing and organization**
>   - We thank you for the remark and will correct the typos and the missing reference. We will improve the structure by better separating the dataset and evaluation sections. We will also better integrate the real-world validation with the main method section to make the paper more cohesive.
> - **Dataset scope limited to GPT-4**
>   - We believe the dataset scope to be limited not to GPT-4’s capabilities, but to the breadth of open-source scientific protocols on protocols.io. The scientists that manually review the generated pseudocode and pseudofunctions were instructed to check if the pseudocode accurately matches the step-by-step instructions in the original protocols (as in Figure 2), making the capabilities on GPT-4 mostly irrelevant when it comes to the ground truth pesudofunctions and pseudocode.
> - **Evaluating on machine-generated data**
>   - While it is true we evaluate on machine-generated data, we (i) generate the pseudo ground truth on an easier task, and (ii) manually ensure it correctly reflects the original protocol. Although we agree the generated pseudocode might contain a bias towards the generations of GPT-4, we present a framework that can be applied to any LLM, and is the first, to our knowledge, automatic evaluation method of planning tasks described in plain text that does not require a simulator. Furthermore, this is a significant step forward from prior works (Wang et al., 2023) that evaluate GPT-4 responses using GPT-4, by simply asking it to evaluate a response given a ground-truth.
> - **Dataset size**
>   -  We have increased the number of manually verified protocols to 100 and we are committed to continue increasing the size of the dataset. Furthermore, to improve the robustness of the findings, we report mean and standard deviations over 5 generation runs for the tasks of protocol generation and next step prediction below. We will report these in the camera ready version of the paper.
>
>   Protocol Generation
>
>   | Model   | Shuffle | Feedback | Precision       | Recall        | $\mathcal{L}_{dn}$ $\downarrow$ | Args Precision | Args Recall   | SciBERTScore | BLEU            |
>   |---------|---------|----------|-----------------|---------------|---------------------------------|----------------|---------------|--------------|-----------------|
>   | GPT-3.5 |  x      |   x      |  93.4&pm;0.9    |  89.9&pm;0.6  | 0.498&pm;0.036                  |  72.7&pm;0.8   |  91.4&pm;1.5  | 82.7&pm;0.6  | 0.121&pm;0.005  |
>   | GPT-3.5 |  x      |   v      |  93.3&pm;1.0    |  91.1&pm;1.1  | 0.505&pm;0.159                  |  73.1&pm;1.6   |  88.1&pm;1.9  | 82.8&pm;0.6  | 0.117&pm;0.006  |
>   | GPT-3.5 |  v      |   x      |  91.8&pm;0.8    |  85.9&pm;2.8  | 0.945&pm;0.055                  |  72.9&pm;1.4   |  89.1&pm;2.2  | 81.8&pm;0.2  | 0.102&pm;0.003  |
>   | GPT-3.5 |  v      |   v      |  92.5&pm;0.3    |  86.1&pm;1.6  | 0.884&pm;0.045                  |  73.2&pm;1.3   |  87.3&pm;3.5  | 82.3&pm;0.4  | 0.102&pm;0.009  |
>   | GPT-4   |  x      |   x      |  91.9&pm;0.9    |  90.8&pm;0.9  | 0.396&pm;0.046                  |  72.2&pm;0.8   |  94.7&pm;1.4  | 82.6&pm;0.2  | 0.124&pm;0.006  |
>   | GPT-4   |  x      |   v      |  92.5&pm;0.3    |  90.1&pm;0.3  | 0.438&pm;0.412                  |  72.0&pm;0.3   |  93.3&pm;1.0  | 82.7&pm;0.3  | 0.112&pm;0.005  |
>   | GPT-4   |  v      |   x      |  92.6&pm;0.9    |  87.7&pm;0.9  | 0.722&pm;0.311                  |  72.2&pm;0.3   |  94.6&pm;1.8  | 82.7&pm;0.4  | 0.113&pm;0.004  |
>   | GPT-4   |  v      |   v      |  92.8&pm;1.0    |  86.6&pm;0.3  | 0.685&pm;0.178                  |  73.7&pm;0.7   |  93.4&pm;2.0  | 82.5&pm;0.7  | 0.108&pm;0.004  |
>
>   Next Step Prediction
>
>   | Model   | Shuffle |  Func Accuracy  | Args Precision | Args Recall   | SciBERTScore | BLEU            |
>   |---------|---------|-----------------|----------------|---------------|--------------|-----------------|
>   | GPT-3.5 |  x      |  65.8&pm;1.3    |  97.7&pm;0.5   |  94.7&pm;0.5  | 88.5&pm;0.5  | 0.363&pm;0.012  |
>   | GPT-3.5 |  v      |  36.1&pm;1.6    |  97.1&pm;1.2   |  95.1&pm;1.0  | 88.6&pm;0.5  | 0.384&pm;0.028  |
>   | GPT-4   |  x      |  70.6&pm;0.4    |  97.1&pm;0.5   |  94.9&pm;0.6  | 87.9&pm;0.5  | 0.351&pm;0.017  |
>   | GPT-4   |  v      |  57.0&pm;0.8    |  97.1&pm;0.4   |  94.7&pm;0.8  | 88.5&pm;0.6  | 0.363&pm;0.025  |

---

### Official Review · Reviewer_9Mgr · 2023-08-04

**Soundness:** 5

**Excitement:**

5: Transformative: This paper is likely to change its subfield or computational linguistics broadly. It should be considered for a best paper award. This paper changes the current understanding of some phenomenon, shows a widely held practice to be erroneous in someway, enables a promising direction of research for a (broad or narrow) topic, or creates an exciting new technique.

**Paper Topic And Main Contributions:**

The paper "BioPlanner: Automatic Evaluation of LLMs on Protocol Planning in Biology" propose a novel method for self-evaluation of LLMs on the task of automated experimental protocol generation. For this, the authors introduce a new dataset, BioProt, which maps natural language protocols to a pseudocode-based protocol representation. They create this dataset by a series of GPT-4 prompts that transform a gold-standard protocol and a subsequent manual verification and correction step.  The pseudocode-based representation can then be used as a generation target. This makes evaluation easier because the output space is much more constrained than for natural language. The authors propose three different downstream tasks on BioProt: Predicting the next step in a protocol, generating a protocol from scratch, and retrieving the set of needed steps from a pool of pseudo-functions. The authors evaluate GPT-3.5 and GPT-4 on these three downstream tasks and find that GPT-4 generally outperforms GPT-3.5. As a further contribution, the authors extend an LLM with search capabilities and let it design two experimental protocols. They validate the protocol by executing it in the wetlab and find that it leads to the desired outcome.

**Questions For The Authors:**

- Which LLM was used in the Real-world validation?

**Reasons To Accept:**

- The paper tackles the important task of supporting experimental scientists by assisting with protocol generation, which can be cumbersome and error-prone.
- The idea of using an easy task with full information first and then evaluating in a scenario with restricted information is a creative solution for reliable self-evaluation. This might be of interest in completely different tasks than science automation/assistance.
- The authors release a new dataset with three novel tasks for protocol generation.
- The real-world validation impressively showcases that LLMs are ready to assist experimental scientists in protocol generation.
- The paper is well-written and easy to follow

**Reasons To Reject:**

- The real-world validation seems to be a bit methodologically disconnected from other experiments. Why did the authors opt to use the toolformer component only in this experiment and not in the other evaluation experiments?
- The authors evaluate only closed-source LLMs which might lead to severe problems regarding replicability.

**Reproducibility:**

4: Could mostly reproduce the results, but there may be some variation because of sample variance or minor variations in their interpretation of the protocol or method.

**Reviewer Confidence:**

3: Pretty sure, but there's a chance I missed something. Although I have a good feel for this area in general, I did not carefully check the paper's details, e.g., the math, experimental design, or novelty.

---

> ### Author Rebuttal · Authors · 2023-08-28
>
> We thank you for the encouraging feedback. We are glad you believe that there is utility in using our framework for self-evaluation beyond this task, and that it can be of interest in different fields.
>
> - **Additional models**
>   - We evaluate an additional, open-sourced model -– Llama-2 7B. We hope this improves the reproducibility of our work. Please find the results in the response to reviewer Lo15
> - **Real-world validation**
>   - We use real-world validation as a case study and to measure to what extent the proposed dataset and framework can be used beyond model evaluation. We opt for an agent with access to tools, such that it can retrieve protocols that are found useful, and use their pseudofunctions – as opposed to generating psueodofunctions from scratch. This ensures the used pseudofunctions are actual atomic actions that are executable in a lab, as opposed to generating them from scratch. Note that for good performance in this real-world validation, the LLM needs to be able to (1) find relevant psueodofunctions from other protocols, and (2) generate correct pseudocode, both of which are tasks we propose and evaluate on. We used GPT-4 throughout this evaluation. We will make both the purpose of real-world validation and the model architecture clear in the paper.

---

### Official Review · Reviewer_Lo15 · 2023-08-05

**Typos Grammar Style And Presentation Improvements:** Broken reference in L449.
**Soundness:** 3

**Excitement:**

3: Ambivalent: It has merits (e.g., it reports state-of-the-art results, the idea is nice), but there are key weaknesses (e.g., it describes incremental work), and it can significantly benefit from another round of revision. However, I won't object to accepting it if my co-reviewers champion it.

**Paper Topic And Main Contributions:**

This work proposes a new task involving language models generating biology experiment/process protocols based on steps expressed in pseudocode. The data is taken from the Protocols.io website, the data creation is performed with the help of GPT-4, and the authors evaluate GPT-3.5 and GPT-4 on it.

**Reasons To Accept:**

The authors present a new dataset and task-type that will be useful to evaluate models on. The data creation procedure is well-described and thought out, with a range of metrics to evaluate on. The authors also include an actual biology experiment conducted based on model outputs.

**Reasons To Reject:**

The paper currently only evaluate on GPT-3.5 and GPT-4, which is a fairly limited slice of model, and moreover there is a clear overlap with GPT-4 also being used to construct parts of the dataset. (The authors do highlight that prior work has noted this). This may artificially boost the GPT-4 results. Ideally the models should also evaluate on models outside the OpenAI family, particularly open-access models.

**Reproducibility:**

3: Could reproduce the results with some difficulty. The settings of parameters are underspecified or subjectively determined; the training/evaluation data are not widely available.

**Reviewer Confidence:**

2: Willing to defend my evaluation, but it is fairly likely that I missed some details, didn't understand some central points, or can't be sure about the novelty of the work.

---

> ### Author Rebuttal · Authors · 2023-08-28
>
> We thank the reviewer for their thoughtful review and useful feedback.
> - **Additional models**
>   - We agree that not evaluating open-source models is a limitation. We thus evaluate Llama-2 7B on BioProt. Please see the results below for LLama-2 7B on the protocol generation and function selection tasks. We will replicate all other experiments on Llama for the camera ready version of the paper.
>
>   Protocol Generation:
>
>   | Model     | Shuffle | Feedback | Precision | Recall | $\mathcal{L}_{dn}$ $\downarrow$ | Arg Precision | Arg Recall | SciBERTScore | BLEU   |
>   |-----------|---------|----------|-----------|--------|---------------------------------|---------------|------------|--------------|--------|
>   | Llama2-7B |  x      |   x      |  83.6     |  49.8  | 0.74                            |  76.2         |  41.4      | 79.8         | 0.048  |
>   | Llama2-7B |  x      |   v      |  81.0     |  45.9  | 0.82                            |  70.4         |  42.9      | 80.4         | 0.050  |
>   | Llama2-7B |  v      |   x      |  82.2     |  45.1  | 0.63                            |  70.7         |  43.8      | 81.3         | 0.051  |
>   | Llama2-7B |  v      |   v      |  78.5     |  30.4  | 0.56                            |  73.0         |  51.4      | 81.1         | 0.047  |
>
>   Function Selection
>
>   | Model     | Neighbourhood| Precision | Recall |
>   |-----------|--------------|-----------|--------|
>   | Llama2-7B |  Nearest     | 26.1      |  57.5  |
>   | Llama2-7B |  Random      | 28.1      |  56.3  |
>
> - **Artificially boosted GPT-4 results**
>   - The aim of our work is not to show that one model is superior to another when it comes to scientific protocol generation. The aim is instead to propose a framework for evaluating LLMs on planning tasks, where there is no automatic way of evaluating the response. We will make sure this is more explicit in the abstract of the paper. Furthermore, the concern that GPT-4 results are artificially boosted is mostly valid for prior works that use GPT-4 as an evaluator of GPT-4 responses. We however, improve on this by automatically creating pseudo-ground-truth responses on an easy task, with additional context, and then evaluating the LLM’s responses using closed-form evaluation metrics. While we agree there is a possibility of leakage because of the use of pseudo-ground-truths generated by GPT-4, we mitigate this by having scientists manually inspect the generated pseudofuctions and pseudocode to make sure it is correct. We find that 59% of the responses are entirely correct (now that we have increased the dataset size to 100 protocols), and the ones that are incorrect are manually fixed. We will add a discussion of how the ground truth responses might be biased to GPT-4. However, these ground-truth responses can be generated by any LLM, as long as they are found to be correct.
> - **Reproducibility**
>   - With evaluations of open-source models, full documentation of our metrics, and all prompts and data released in the supplementary material, we believe our work is reproducible.

---

### Meta-Review · Area_Chair_nsat · 2023-09-17

**Recommendation:** 4

**Metareview:**

This work investigates a unique and interesting application (supporting experimental scientists by assisting with protocol generation) and presents a new evaluation set and method for self-evaluation. Though there are valid concerns among reviewers that evaluating GPT-4 with GPT-4 will have misleading biases (and also questions about training set representation), these experiments and resources are likely to prove insightful to practitioners.

---

### Decision · Program_Chairs · 2023-10-07

**Decision:**

Accept-Main

**Comment:**

This work investigates a unique and interesting application (supporting experimental scientists by assisting with protocol generation) and presents a new evaluation set and method for self-evaluation. Though there are valid concerns among reviewers that evaluating GPT-4 with GPT-4 will have misleading biases (and also questions about training set representation), these experiments and resources are likely to prove insightful to practitioners.